# Impact of COVID-19 Confinement on Mental Health in Youth and Vulnerable Populations: An Extensive Narrative Review

Manuel Reiriz [1,*], Macarena Donoso-González [2], Benjamín Rodríguez-Expósito [3], Sara Uceda [1] and Ana Isabel Beltrán-Velasco [4,*]

1   BRABE Group, Department of Psychology, Faculty of Life and Natural Sciences, University of Nebrija, 28248 Madrid, Spain
2   Department of Theory of Education and Social Pedagogy, Faculty of Education, Universidad Nacional de Educación a Distancia, 28040 Madrid, Spain
3   Education Department, Faculty of Languages and Education, University of Nebrija, 28240 Madrid, Spain
4   Department of Psychology, Faculty of Life and Natural Sciences, University of Nebrija, C/del Hostal, 28248 Madrid, Spain
*   Correspondence: mreiriz@nebrija.es (M.R.); abeltranv@nebrija.es (A.I.B.-V.)

**Abstract:** The objective of this narrative review is to analyze the impact of COVID-19 on the mental health of particularly vulnerable groups. This information will allow a better understanding of the determining factors that influence the appearance and/or maintenance of mood disorders. To achieve the main objective of this study, a critical review was carried out in which primary sources such as scientific articles, secondary sources such as databases, and other appropriate reference indexes were considered. The results indicated that there was an increase in the diagnosis of mood disorders and the use of medication associated with these disorders, mainly during the period of reclusion that was declared worldwide in March 2020. In addition, risk factors such as loneliness, a lack of resilience, and a lack of adequate coping strategies negatively impacted these groups. The future consequences of this may be reflected over many years thereafter, and it is important that all data obtained from this point forward be considered by mental health professionals and the general population. This review can be a starting point for looking directly at the most vulnerable populations and considering both the resources available to them and the possible aftermath of a traumatic period in everyone's lives.

**Keywords:** COVID-19; mood disorders; medication use; information; confinement





## 1. Introduction

People had to adapt to an exceptional situation during the confinement decreed by the global health authorities after the emergence of COVID-19. This period meant a radical change in the habits of everyone, from the youngest to the oldest, which had a high impact on different areas of life. Among the most affected, we can undoubtedly say at this point that it has been mental health [1–6].

Several sources indicate that mental health problems have increased considerably during the COVID-19 pandemic, with a significant increase in symptoms associated with depression and anxiety disorders [7]. This also includes an increment in levels of distress, even among the younger population, with these symptoms being disabling and causing great discomfort and suffering in individuals. On the other hand, it is also important to know that these effects still have relevance today [8].

Confinement had a high impact at the psychological level, which was caused by the political and health measures taken by different governments with the aim of slowing down the advance of the disease [9]. These measures first proposed home confinement which, on many occasions, was associated with loneliness or, even worse, made it easier for people to have access to causing distress to the people they were living with [10].

This extreme situation, which caused a change in the lifestyle habits of the entire population, was not determined with the possible consequences on people's mental health, but to prevent the spread of a disease that caused thousands of deaths. However, another pandemic appeared at the end of the first one, and it is that of mental disorders whose diagnosis is associated with confinement [9].

Thus, the data that have been obtained in different studies have shown the need to record the negative effects of the pandemic on the mental health of the world population, specifically during the period of confinement, also taking into account factors that correlate with the appearance of and an increase in mental illnesses such as age, gender, or the vulnerability of certain groups, which have predisposed them to a higher risk of presenting with symptoms associated with mood disorders [11].

Any epidemic or natural catastrophe is a stressful event, whose medium- and long-term effects can be present in different ways, particularly at the cognitive and psychological level as this type of event places people in a situation of psychological vulnerability. These circumstances and the events related to them caused a significant increase in deaths worldwide, involving the loss of loved ones, friends, co-workers, etc., and increasing the level of uncertainty and fear that can trigger the emergence of certain negative feelings and misperceptions of reality. However, it also increases the need to cope with these experiences and the associated traumatic events, as well as the presence of a nearby danger that can directly affect the person. This causes emotions such as uncertainty, uneasiness, and fear. All of them put the person on alert and predispose him/her to develop anxious–depressive pathologies [12].

Social isolation is associated with feelings of loneliness and general discomfort that have a great psychological burden but also affect the physiological level. In fact, studies in this line indicate that a percentage of close to 85% of the participants reported sleep disturbances, both in terms of falling asleep and during the night, with interrupted sleep, light sleep, early awakening, etc. [13]. Other participants indicated presenting with muscular pain, recurrent headaches, difficulties in attention and concentration, panic attacks, and feelings of aggressiveness and irritability, which are associated with muscular tension and the presence of dysfunctional anxious symptoms [14]. In addition, these individuals had feelings of insecurity, a fear of death, both their own and that of loved ones, a fear of COVID-19 infection, and even thoughts of death, self-harm, and, in the most severe cases, suicidal ideation [15].

Another important factor in this scenario was social networks and communication channels. At this time, news about the disease was published very quickly, with hardly any time to confirm it, which caused an avalanche of information for the entire population. This type of information fostered an increase in fear or embarrassment in the face of the impossibility of being able to implement protective actions or behaviors. In addition, it provoked a feeling of absolute helplessness in the face of a silent disease that was spreading without distinction and with a daily increase in the number of deaths, which made the fear even greater [16]. In addition, social networks were filled with people sharing their experiences, their ideas, and their subjective and individual perceptions of what they were experiencing, and this provoked a feeling of extreme fear in certain groups, who made attributions and inferences based on opinions that may or may not have a scientific basis [17]. Although on many occasions this type of testimony was made with the intention of helping other people, the reality is that it provoked a greater sense of fear, hopelessness, and uncertainty about the future and with it, the appearance of avoidance behaviors and emotional and psychological problems [18].

A comprehensive review of these elements and how they have impacted individuals is important to determine the health status of the population two years after the onset of the pandemic. These data are essential to determine the best way to help affected individuals, what resources are currently needed to reduce the presence of these anxious–depressive symptoms, reduce the pharmacology associated with these disorders, and provide appropriate tools to individuals to alleviate the suffering and distress associated

with these pathologies. Thus, the objective of this narrative review is to analyze the impact of COVID-19 on the mental health of particularly vulnerable groups.

## 2. Increased Diagnosis of Mood Disorders

The COVID-19 pandemic brought populations to an environment and emotional situation that resulted in a significant increase in psychological disorders such as depression, stress, irritability, mood swings, emotional distress, fear, and anxiety [19,20]. In this line, the experiences lived by people during the months that the pandemic lasted (the pathogenicity of the virus, the resulting high mortality rate, or the quarantine experience) affected the mental health of the general public, especially healthcare workers, patients with mental illnesses, or infected patients. Because of this, it is important to note that there are several groups that could be called vulnerable, with the groups mainly being those who remained alone during confinement, those who already had a clinical diagnosis of physical or mental pathologies, and those who were dependent or living with dependent people [21–23].

Due to this, there was an increase in the diagnosis of mood disorders in different countries around the world compared to the period prior to the pandemic. Thus, a study conducted by McGinty et al. (2020) in the USA reported an increase in psychological distress in April 2020 (one month after the USA quarantine) compared with in 2018 in adults. In this sense, in April 2020, 13.6% of this population showed symptoms of psychological distress compared to 3.9% reported in 2018, showing that the subgroups examined in April 2020 had the lowest prevalence of serious psychological distress in adults aged 55 years or older [24]. The same results were obtained by Pierce et al. (2020) in the population of the UK, where the prevalence of mental distress raised from 18.9% in 2018–2019 to 27.3% in April 2020 (one month after the UK quarantine). In addition, the diagnosis of anxiety in the general population measured with GHQ-12 also increased in that period from 11.5% in 2018–2019 to 12.6% in April 2020 [25]. On the other hand, several studies carried out in China showed that the incidence in the general public of mood disorders such as depression and anxiety increased due to the COVID-19 pandemic. Thus, before February 2020, the prevalence of these mood disorders was 24% and 33% for anxiety and depression, respectively [26,27], with these values increasing after this date by reaching 44% and 62% for anxiety and depression, respectively. This information can be seen in Figure 1 [20]. Likewise, according to an observational study conducted by Wang et al. (2020) during the initial phase of COVID-19 in China, 30% and 37% of the general public suffered from depression and anxiety, respectively [28].

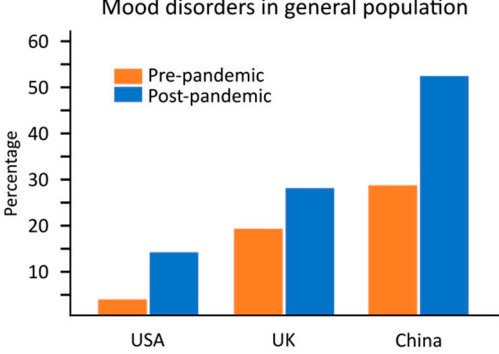

**Figure 1.** Increase in the diagnosis of mood disorders pre- and post-pandemic on different continents. Own elaboration.

One substantial fact related to the effect of the pandemic on the incidence of mood disorders in the general population was that the female gender is at a higher risk of depression symptoms, anxiety, and post-traumatic stress disorder (PTSD), as shown by the results of several epidemiological studies [29–31]. In this sense, during the pandemic, the prevalence in women of depression, stress disorders, and anxiety was higher than in men [32–34].

Healthcare professionals had to manage against the virus and face stressful situations such as making decisions of moral responsibility and those associated with the care of their patients. They also had to cope with increased emotional stress due to high exposure to a deadly and unknown disease and the uncertainty and fear of catching or infecting their significant others. In addition, there was a lack of support from public administrations during the pandemic, which could be involved in the increased risk of mood disorders in this group [35,36]. Under these circumstances, it would be expected that the incidence of mood disorders in healthcare workers would be higher than in the general population, but the evidence found by Deng et al. (2021) in China showed a different tendency among healthcare workers and the general public before and after the peak. They observed that before the peak of the epidemic (8 February), the incidences of anxiety and depression in healthcare workers were 40% and 31%, respectively, while after the peak the incidence of both mood disorders was 22% [37]. In this line, the incidence related to mood disorders among healthcare workers declined after the peak compared to before, while in the general group, a rising tendency of the incidence of mood disorders was detected [37]. This result was confirmed by several studies in other countries which determined that during the pandemic the prevalence rates of anxiety and depression in healthcare workers were between 26–23.2% and between 12–22.8%, respectively [38,39]. However, the incidence found in healthcare workers due to the pandemic remains high.

On the other hand, in vulnerable groups, such as people with mental illnesses, the emotional responses generated by the unprecedented stressor that the pandemic and its management could be more important than in the general population [40,41]. In this sense, different studies have found that the levels of mood disorders such as symptoms of loneliness, anxiety, depression, and worry increased systematically in people with pre-existing psychiatric conditions during the pandemic compared with the pre-COVID-19 situation [40–42]. This fact could be explained because in people with mental illnesses, factors such as small social networks or financial instability are common, and the quarantine experience could have aggravated this situation [43].

## 3. Increased Medication Use

Since the pandemic began, there has been a significant increase in diagnoses of depression and anxiety, as well as mixed mood disorders. Studies conducted in the following months indicated an increase of up to 25% in these disorders, as well as in their associated symptomatology [44].

A significant proportion of the general population has been negatively affected by home confinement, which was carried out for more than 100 days in most countries around the world [45]. The impact of this extraordinary state that modified the way of life of all people is related to social isolation, as well as a significant reduction in perceived support through networks of friends, partners, and even family [46]. Other factors that influenced the appearance of mood disorders include the loss of employment, exposure to violent environments in the home, alcohol and other substance use, abusive situations, the presence of other pathologies, etc. [47].

This rather complex scenario has led to an increase in the intake of drugs associated with anxious and depressive disorders, i.e., anxiolytics and antidepressants. Along these lines, several studies have been able to compile data on the use of drugs during and after this time of confinement [48]. During the confinement period, there was a paradoxical effect in which the prescription of anxiolytics, hypnotics, and sedatives in general decreased considerably, probably due to a reduction in health services, as well as to the fear generated in the population of going out into the street and becoming infected. Therefore, this decrease in the prescription of these drugs could be explained by the need to acquire them under medical prescription and by the decrease in access to health services [49]. However, this is not related to the symptomatology of mood disorders, which increased during these months, along with the appearance of emotions such as fear and anxiety, and the increased prevalence of pathologies such as depression and anxiety. Therefore, this seems

to indicate that this decrease in the consumption of anxiolytics is not a good indicator of people's mental health during this period. Even more so knowing that social distancing and the suspension of life habits and social interactions had a precisely predisposing effect on the vulnerability to these mental pathologies [50]. However, after an initial decrease in the prescription of medication, there was a subsequent slight increase, which was not significant, but indicated a return to the patterns prior to the start of confinement period [11].

Regarding antidepressants, an increase in the prescription of these drugs was observed just before the pandemic was declared, and at the beginning of the confinement period there was a decrease in their use [51]. This can be explained by the same reason as in the case of anxiolytics and sedatives, i.e., restrictions on access to health services, as well as an evident decrease in medical visits at a time when the health priority was the devastating reality of the daily number of deaths due to COVID-19. In the months following the end of confinement, no significant increase in the prescription of antidepressants was identified, although it is not possible to determine whether this is due to avoidant healthcare behaviors on the part of mental health patients [52].

However, other studies in this line have shown that young people have reported having consumed anxiolytics and antidepressants. In addition, it has been reported that they consumed melatonin to help them fall asleep, as well as other natural medications such as valerian or Passiflora [53]. These results seem consistent with previous studies indicating that young people frequently consume psychotropic drugs, both prescribed and obtained from family and even friends [54]. In these studies, a tendency of this sector of the population to consume common antidepressants and benzodiazepines, specifically diazepam, can be observed, and this tendency was maintained during confinement [55].

It seems important to be able to determine the consumption of these drugs during the period of confinement to try to identify behavioral patterns in individuals who present symptoms associated with mood disorders. Having these data may help us to predict trends in drug use, as well as the individual determinants that impact the decision to self-medicate.

## 4. Loneliness and Depression

Recently, because of the COVID-19 pandemic, we have been experiencing a significant deterioration in the mental health of our society with relevant psychological consequences. In this line, this unexpected situation has been the trigger for multiple diseases and negative situations that are closely associated with the psychosocial well-being of all people, especially, as recent studies indicate, the most vulnerable population: the young and the elderly [56].

As a result of social isolation, the feeling of loneliness emerges as one of the main elements that has contributed to the deterioration of the psychosocial health of both groups [57]. Thus, due to various limitations and restrictions, contact with partners was drastically reduced, increasing the negative psychological effects associated with loneliness, anxiety, and depression [58]. This disruption of social activities has led to the emergence of new feelings of negativity and psychological pathologies, as well as the deterioration of existing ones in this group of people [59,60]. Stress, anxiety, negative self-perception, homesickness, and a strong feeling of helplessness and loneliness have led to a significant increase in psychological risks and uncommon diseases in this population [61–63]. This has been confirmed by several studies, alluding to worsening feelings of loneliness (19.6% in the population >30 years and 8.9% in the population <50 years), increased levels of distress and hopelessness (33.5% in the population >30 years and 16.7% in the population <50 years), altered levels of stress (38% in the population >30 years and 14.4% in the population <50 years), and anxiety (29.1% in the population >30 years and 15.6% in the population <50 years) [21,64,65].

The breakdown of daily activities in these age groups promoted the appearance of feelings of apathy. These included sports, school, and artistic activities for the younger

ones, and walks, shopping, and meetings with peers for the oldest. In short, activities of an imminently social nature were paralyzed, leading to emotions of loneliness and loss of interest, associated with depression (32.4% in the population >30 years and 18.9% in the population <50 years), which transgressed certain psychosocial limits, giving way to a lack of energy and lethargy, both breeding grounds for depressive processes of a certain depth [64,66]. In addition to these important consequences, it has been shown that during the pandemic, other pathologies appeared and worsened in young people and the elderly: an increase in the youth population in the obese range, changes and disorders associated with eating disorders in adolescents, continued irritability and anger (36.4% in the population >30 years and 12.2% in the population <50 years), difficulty in attentional processes and concentration (25.7% in the population >30 years and 14.4% in the population <50 years), body pain as a consequence of prolonged maintenance of unhealthy postures, cognitive impairment and loss of mobility in the elderly, etc. [67–70].

These determinants related to the health of young people have also given rise to other more serious psychological problems, mainly depressive disorders and feelings associated with loneliness and the absence of social relationships [71]. Thus, the new reality includes an increase in drug use and abuse, addiction to technologies, loss of social skills, disruptive behaviors in the family environment, and an increase in suicidal thoughts or behaviors. This reality has become common in the younger population and significant risks arise in the emergence of mental health disorders [72–74].

Regarding social isolation, other factors have increased depressive and stressful tendencies among young and old people; the loss of family members and other relatives (more than 70% of the population has shown concern about this issue), the loss of their own or their parents' jobs, the fear of death, job uncertainty, the management of new family and work relationships (teleworking), among others, have been the most prominent causes of this population's unease [75,76]. In this sense, we should add more extreme cases, such as those existing in the homes of the youngest, such as abuse and domestic violence, which have caused a collapse in the psychosocial well-being of this vulnerable population.

In short, it seems clear that the adaptive coping of young and elderly people in naturally adverse situations, such as the COVID-19 pandemic, is much lower than that of other population groups. These age groups, as the studies point out, have reported a higher propensity to suffer from loneliness, depressive tendencies, post-traumatic stress disorder, and the occurrence of anxiety [77,78].

## 5. COVID-19 Confinement and Its Impact on Vulnerable Populations

The coronavirus pandemic has had a great impact on different aspects of our lives, such as our physical and mental health. Thus, there are several studies that show how the pandemic, and specifically lockdown, increased anxiety and depression in the general population [28,79–82]. Moreover, this effect on mental health was related to different factors such as the duration of the lockdown period, social distancing, financial problems, and changes in sleep patterns or diet, among others [81]. However, there are some population groups who are more at risk of a possible COVID-19 infection or have a worse prognosis because of their vulnerable situation. Among these groups, we can highlight the homeless, people who suffer from other illnesses or infections, or people with mental health issues. In this section, we revise how the COVID-19 pandemic and lockdown affected the physical and mental health of these populational groups.

### 5.1. COVID-19's Impact on the Homeless

Since the start of the coronavirus pandemic, the impact of COVID-19 has not been the same in all populational groups. To deal with this, many governments implemented different measures to attend to many groups such as elderly people [83], communities from different races, or people with disabilities [84,85]. However, the measures adopted for the homeless (a community that represents 1.6 billion people around the world with, for example, 900,000 people in France alone or 1.7 million in India [86–88]) were few and

late [89,90], showing again how this population group, especially those who have drug addiction or mental health problems, has been considered as a highly marginalized and neglected population worldwide [91,92]. Effectively, the homeless have a higher probability of suffering from mental health problems or drug abuse [93,94]. For example, in Hong Kong, the homeless are more likely to suffer from depressive disorders than the general population [95]. Other studies have shown that the homeless have severe mental disorders (such as schizophrenia and bipolar affective disorder) or intellectual disabilities [96].

Focusing on the COVID-19 pandemic, a study that compared the mental health state in the homeless and the general population before and during the lockdown period concluded that 81.5% of the homeless participants showed lower levels of good mental health compared with the general population both before and after the lockdown period [97]. Thus, the restrictions of the lockdown period directly affected the homeless. For example, in the UK, few beds were offered to the homeless due to social distancing, access to drug support groups was suspended for several months during the lockdown period, and it was more difficult to access food or essential health services. In addition, in some cases, evicting the homeless from urban areas was a measure implemented, increasing their isolation [89,98,99]. Moreover, in these persons, the COVID-19 prevention measures (i.e., social distancing or hygiene) were different than in the general population due to their poor sanitary conditions or the use of shelters among other circumstances. This situation generated an environment that was conducive to a disease epidemic [100,101]. Therefore, the homeless are considered as a vulnerable group for COVID-19 infection and this could have a negative effect on both their mental and physical health [101]. Moreover, there are some additional issues that are unique to people experiencing homelessness, for example, with them being more mobile than the general population, thus making it more difficult to track transition or to treat infection [93].

Therefore, the homeless have a higher probability of COVID-19 infection and the mortality rate in this group is higher than in the general population [101]. Thus, a study conducted by Roederer et al. (2021) analyzed 818 people who were living in different facilities during the first month of the COVID-19 pandemic in France. The study showed how 52% (426) of them were positive in any serological test. These results show higher seropositivity in homeless population compared to the general population, from 9% to 11%, as showed by Le Vu et al. (2020) [102]. Moreover, seropositivity risk factors were most strongly associated with crowded living conditions or less frequent movement during confinement. Indeed, people who never left their place of residence were more likely to be COVID-19-positive and the more crowded the residence was, the higher the probability of being COVID-19-positive. These data show the necessity of implementing adequate housing conditions to these persons, especially those who have comorbidities [103]. However, other studies have shown how confinement could be used for improving the physical and mental health of the homeless. Thus, Thomas et al. (2021) described how an increase in mental health after 4 weeks from the beginning of the confinement period was associated with an increase in physical activity in homeless people [97]. Another study conducted by Martín et al. (2021) in Spain managed to improve the mental health of homeless people through the implementation of a program of serial visits to a shelter for homeless people who were confined to monitor treatment and mental disorders individually; for example, the percentage of people with the main treatment increased significantly (from 58.8% to 82.3%) and the follow-up of this group reduced the psychiatric emergencies that were attended from 23.6% two months before the pandemic to 59% during the pandemic. These studies show how the confinement situation was able to improve the mental health of the homeless using adequate programs [104].

### 5.2. COVID-19's Impact on Other Physical Illnesses

5.2.1. Sexually Transmitted Infections and HIV

The COVID-19 pandemic, especially the confinement period, affected the control of other infections such as sexually transmitted infections or HIV [105–109]. Thus, in

some countries such as Lebanon, there was an increase in the prescription of post-exposure prophylaxis (PEP) of 34% compared with 2019, although in other countries such as Australia, the confinement period caused a huge reduction in PEP prescription (37% in Melbourne and 46% in Sydney). PEP is a therapy which consists of taking a combination of three antiretrovirals for one month after HIV exposure to prevent infection [110], and its use during the pandemic showed how some individuals continued having sexual intercourse during the confinement period [111]. The study carried out in Australia also found a decrease in the number of HIV tests performed (41% in Melbourne and 32% in Sydney) and in the new cases of HIV diagnosed (44% in Melbourne and 47% in Sydney). Similar results were obtained in other countries such as the UK and Spain where the prescription of PEP, the number of HIV tests performed, and the number of new HIV cases diagnosed were reduced as well [112,113]. Thus, although it is probably the case that these effects could be explained by the reduction in sexual intercourse during the confinement period, there were other reasons such as a fear of attending HIV testing services during the pandemic due to the possibility of COVID-19 infection [114,115]. This situation along with the substantial reduction in HIV prevention measures during the COVID-19 confinement period could have resulted in increased HIV transmission [116] that could have resulted in a significant reduction in the mental health of these persons. Thus, HIV infection produces a higher probability of suffering from mental health problems ranging from acute stress reactions to neurocognitive disorders [117,118]. Moreover, in countries such as Uganda, people living with HIV increased their depression levels during the confinement period and this change was associated to antiretroviral non-adherence increasing the risk of transmission [119].

Another HIV prevention strategy that was affected by the coronavirus pandemic was pre-exposure prophylaxis (PrEP) which consists of daily or even driven usage of an antiretroviral before sexual intercourse to avoid the risk of HIV infection [120]. The use of this prevention measure increased during the pandemic, demonstrating it to be effective to protect against HIV infection and to have a huge impact on mental health. Thus, [121,122] a recent study demonstrated how the use of PrEP reduced anxiety and increased sexual satisfaction in men who have homosexual sex in Spain [123]. The impacts of COVID-19 on PrEP use are unclear. Thus, for example, in countries such as South Africa, France, Belgium, the USA, Wales, or the Netherland, the access to PrEP was reduced [124–133] while other authors found an increase in PrEP use, for example, in the Netherland [134]. In other countries such as Australia, the use of PrEP remained more or less stable during the COVID-19 pandemic [115,127]. It is important to note that the use of PrEP as a strategy includes not only access to treatment but also 3-monthly physician visits for HIV and renal function testing and counseling [135–137]. Thus, the change in access to the use of PrEP during the coronavirus pandemic and all the of support associated with it, joined to the impact of this strategy in the anxiety level and sexual satisfaction, could have an impact on the mental health of the people that use this strategy.

### 5.2.2. Cancer

Another disease treatment that was strongly affected by the COVID-19 pandemic was cancer. Thus, the pandemic and the confinement period produced a huge impact on diagnosis and treatment in patients with cancer. For example, in a study conducted with 20.006 participants with cancer in 61 countries, the surgery to cure it was affected by the confinement measures. Specifically, light restrictions were associated with a 0.6% non-operation rate, moderate confinement with a 5.5% rate, and full lockdowns with a 15.0% rate [138]. In addition, the COVID-19 pandemic and the confinement period have had a huge impact on people with cancer's mental health. Thus, although loneliness during the pandemic seems to affect both people with and without cancer (in fact, loneliness has been considered as an independent risk factor with a similar effect to smoke 15 cigarettes per day in both cancer and non-cancer people [139–143]), people with any medical condition as such cancer were more likely to be isolated during the COVID-19 pandemic due to the high risk of severe complications [144]. Therefore, social isolation may contribute to

increasing loneliness in people with cancer (or people who separated by it) during the pandemic [14]. Thus, other studies have shown how patients with cancer or survivors felt more loneliness during the pandemic [145,146] and had worse mental health with more depression, anxiety, and stress compared with their family members [147]. Other authors found an increase during the pandemic in anxiety, stress, depression, sleep disorders, or cognitive impairments in patients with cancer (gastric cancer) [148,149].

### 5.2.3. Other Mental Health Issues

Finally, another population group that has suffered specially from the effects of the coronavirus pandemic and the confinement period is people with mental illnesses. In Spain, Solé et al. (2021) evaluated the differences between this population and community controls. They found how community controls used more adaptative strategies (such as following a routine, physical exercise, a balanced diet etc.) than people with mental health issues. Moreover, people with mental illnesses showed more anxiety, depression, weight gain, sleep changes, and tobacco consumption than the community control. Thus, the authors concluded that "confinement had a higher psychological impact in individuals with a psychiatric illness" [150].

Another group affected by the COVID-19 pandemic and who were studied through a survey made for parents or careers were children and young adults with physical and/or intellectual disabilities. The authors found that 90% of them suffered a negative impact on their mental health during the pandemic and the confinement period (with poorer behavior, mood, fitness, and social and learning regression). Most of the people surveyed highlighted problems in accessing specialist facilities, therapies, and equipment and showed concern for their future physical and mental health [151].

Finally, the pandemic has affected other aspects that are related to mental health such as alcohol consumption patterns or changes in our eating behavior. Thus, the pandemic has significatively increased alcohol use. This effect has increased alcohol emergencies; with the COVID-19 infection risk being this negative, its effects are higher in vulnerable groups such as people with co-morbid mental health problems [152].

## 6. Impact of Information during Confinement and the Search for Mental-Health-Related Indicators during Confinement

The coronavirus pandemic has had a huge impact on our lives and from the beginning, there was a lot of information about the virus and the effect of it public health and the economy. As soon as the virus appeared, the information associated with this virus became trending online content and many bloggers, YouTube users, or people sending information via WhatsApp, Facebook, Instagram, or Twitter have appeared [153]. However, this information was not always correct, and the World Health Organization sent out an alert about the existence of an "infodemic", described as "an overabundance of information, some accurate and some not, that makes it hard for people to find trustworthy sources and reliable guidance when they need it" [154]. According to the WHO, the infodemic generates fear and panic due to unchecked mind-boggling rumors, flamboyant news propaganda, and sensationalism [155]. The infodemic was possible for two main reasons, first, because nowadays access to information is easier than ever before [156]. In this sense, in the USA, 89% of young people (13–17 years) had their own smartphone in 2018, with this figure more than doubling over a 6-year period [157]. Moreover, due to the confinement period, people had more time to spend on the Internet and even preferred online over face-to-face channels for entertainment, communication purposes, learning social rules, recreation, or to be informed [158]. Thus, through the analysis of social media, a study conducted by Islam et al. (2020) identified 2311 case of infodemics in more than 87 countries. Most of them (89%) were rumors, conspiracy theories (78%), or stigma (3.5%). In fact, social media has been critical for the genesis of anti-Chinese sentiment around the globe during this pandemic [159]. Moreover, these reports were not uniform at the time but there were three waves between January 21 and February 13 [160].

Sometimes, infodemics directly affected our physical health; for example, in Iran, more than 100 persons died from alcohol poisoning due to the belief that alcohol could kill the virus [161]. Another example was with the access to hydroxychloroquine; many people have used it for decades due to its antimicrobial and immunomodulatory properties; during the COVID-19 pandemic, there was a supply problem due to hydroxychloroquine being proposed as a cure or preventive tool against the coronavirus [162]. In addition, infodemics affected our mental health too. In fact, previous studies have associated indirect exposition to mass trauma using social media with an increase in the symptoms of post-traumatic stress disorder [163]. Thus, some authors found mental health problems related to social media exposure during the COVID-19 pandemic. Specifically, the increase in the use of social media exposure during the outbreak was positively associated with the presence of anxiety, depression, or both [26]. Thus, infodemics can generate alternatives truths, confuse people, generate anxiety, and make people prone to suspicious, xenophobic, or even psychotic and extreme behaviors [164]. There are many studies that have analyzed the effect of infodemics in mental health. For example, Delgado et al. (2021) through a scoping review, concluded that the main effects of infodemics on adult and elderly mental health were anxiety, depression, and stress. Moreover, this study concluded that young adults and females were the populational groups more affected [165]. Thus, different studies are in the same direction; for example Fhon et al. (2022) analyzed the effect of infodemics on elderly people and found how exposition to information during the pandemic in this population was associated with anxiety, stress, and depressive symptoms [166]. Another study showed how 61% of the 1270 persons who participated presented with clinically relevant anxiety and half of those who presented with anxiety had mild to severe sleep disturbance as well [167]. Another author found other effects on mental health evoked by infodemics such as phobias, panic spells, obsession, irritability, delusions of having symptoms similar to COVID-19, and other paranoid ideas [168,169].

To deal with infodemics, some authors have created a list of recommendation. Thus, some of these recommendations are to not associate the infection with any race or ethnicity, try to be emphatic, share information that creates a positive atmosphere (such as altruistic behavior), acknowledge the work of all the people that fight against the virus (police, medical staff, etc.), authenticate the information regarding the disease, or control the time we spend reading news about the pandemic [156].

The prevalence of mental health problems resulting from the COVID-19 pandemic has progressively increased the search for information and indicators related to mental health [170,171]. This is confirmed by several studies that have investigated searches related to psychological processes that have appeared during the pandemic [172,173]. Terms such as anxiety, stress, depression, and insomnia, among others, were the main focus of these investigations [174–176]. These indicators showed clearly higher values after the start of the pandemic was declared by the WHO.

The term "anxiety" was the one that registered the most significant increase in the searches performed by the subjects. The pandemic and the resulting quarantine, with its suspension of work, social activities, etc., clearly caused an increase in the number of people suffering from this psychological process [177,178]. Thus, the search for information, help, advice, etc., in virtual environments has been a constant from March 2020 to the present, with a particular spike in the worst months of the pandemic (March–July 2020) and in the months related to the war in Russia and Ukraine (mainly at the beginning of 2022) [179,180].

On the other hand, terms such as "stress" or "depression" have also seen an increase in the percentage of searches from the first quarter of 2020 to the present, reinforcing the idea that the health scare has been a warning for the mental health of the population. Moreover, the pathologies associated with these indicators, such as "insomnia" and "headaches", among others, have also presented alarming scores, with them being higher for the term "headache", indicating the strong congestion related to mental health among the subjects who initiated these searches [181–183].

Likewise, searches related to mental health professionals, psychologists, and psychiatrists have also seen their percentages increase, with them maintaining a steady rise since the beginning of the second half of 2020, with higher scores for the term psychiatrist, and reaching their highest peak at the beginning of the year 2022 [184]. The same happens with the search for drugs usually recommended medically for the treatment of anxiety and depression; thus, when searching for four of the most used drugs, "lorazepam", "diazepam", "alprazolam", and "clonazepam", we can observe a steady increase in recent years, with the first two obtaining the highest scores. This information can be seen in Figure 2 [185–187].

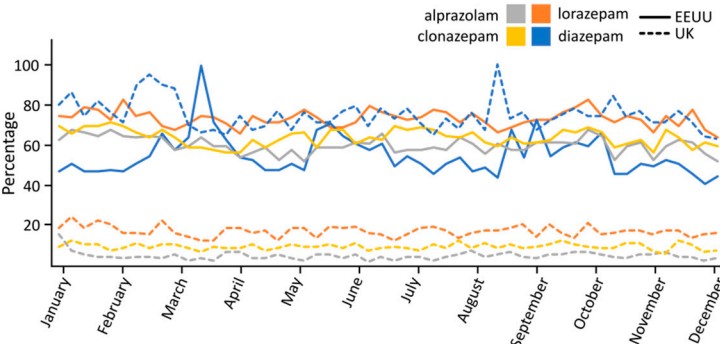

**Figure 2.** Tendency to search for the terms "lorazepam", "diazepam", "alprazolam", and "clonazepam". Own elaboration.

## 7. Individual Resilience Tools in the Face of Confinement

Resilience is a term that has been associated with an adaptation to adversity, which was relevant to face the uncertain situations generated by the COVID-19 pandemic [188,189]. Specifically, individual who are resilient are characterized by a great number of factors and personal skills, such as a sense of hope and safety, calm thinking, or connectedness [190–192]. In this sense, various coping strategies, such as physical activity, going outdoors more often, sleep, prayer, and social support have been associated with increased resilience during lockdown [193]. Thus, a study conducted by Killgore et al. (2020) analyzed how these factors contributed to the resilience of people in the United States during the beginning of the confinement period. In this research, the emotional health, hours of sleep, and their daily activities were studied as well. The results revealed that people with higher family and social support, more minutes of physical exercise daily, more hours of outdoor time in the sun per week, and greater frequency of prayer showed sustained mental health during the confinement period. Therefore, social support from family and friends, exposure to the sunlight each day, and daily activities such as physical activities were associated with greater resilience [193]. Other factors that are related to resilience are a positive appraisal style and good stress response recovery, as shown in the research conducted by Veer et al. (2021) who found that these factors improved the appraisal of specific threat dimensions, such as threat probability or coping potential, which are the dominant stressors in the COVID-19 pandemic situation [194].

Related to the role of physical activity (PA) with resilience factors, Carriedo et al. (2020) carried out research on adults from Spain to establish the effects of physical activity on resilience factors such as the locus of control (the degree of control a person feels he or she has over his or her own behavior), self-efficacy, and optimism during the COVID-19 pandemic. The results revealed that the participants with intensive physical activity were associated with high levels of locus of control, optimism, and self-efficacy. In other words, people who practiced PA regularly during the first week of confinement were more likely to cope better with the stressful emotional situation that they experienced [195]. This is consistent with the evidence found by Borrega-Mouquinho et al. (2021) which

demonstrated that regular physical activity at home improved resilience during the COVID-19 home confinement period [196].

Another aspect involved in individual resilience is prosocial behaviors. In this sense, a study conducted by Esparza-Reig et al. (2022) proved the existence of a direct relationship between prosocial behaviors and resilience in confined persons residing in Spain during the COVID-19 pandemic. Thus, the results obtained showed a direct and positive correlation between resilience and prosocial behaviors, demonstrating how social support received during the conditions of being housebound, both physically with people living at home and remotely through video calls, is directly related to people's resilience [197–199].

To conclude, during the COVID-19 home confinement period, factors such as social contact, physical exercise, time spent outdoors, or prayer helped people to cope with the stressful pandemic conditions. In particular, physical activity and prosocial behaviors were able to positively enhance people's resilience, demonstrating how these daily activities constituted important emotional support during the stressful situation that they experienced.

## 8. Conclusions

Mental health before the pandemic was an area of concern in young people, with a prevalence of 20% for mental disorders, with some sequelae that can become chronic and persist throughout life. Since the pandemic, the mental health of children and young people has deteriorated, with an increase of up to 47% in the diagnoses of mental illness. This has been especially relevant during confinement, a time when all people around the world had their normal habits modified and faced forced confinement for more than three months. Moreover, this confinement has been especially hard for young people belonging to certain vulnerable populations and it has had a greater impact on previously existing pathologies and on those that have appeared because of this period.

This traumatic event has had a significant outcome on the mental health of our young people, and mental health professionals may well have to deal with the consequences for many years to come. Precisely because of this, this review can be a way of gathering relevant information for future research to evaluate the current post-pandemic situation and to be able to make proposals for intervention at this time and ideas for prevention for similar situations in the future, allowing the most vulnerable to equip themselves with adequate tools to deal effectively with this type of supervening circumstances.

### 8.1. Practical Applications

These results can be considered as the basis for the study of certain factors that can be considered risk factors in certain more vulnerable populations. This will allow for a better understanding of the complexity of the pathology presented by individuals and will allow for the planning of more useful and focused interventions for each patient.

### 8.2. Theoretical Implications

From a theoretical point of view, it is important to achieve knowledge of the different elements that have a positive and negative impact on people's mental health. This is of greater importance in certain more vulnerable groups as it allows for the generation of new working concepts and hypotheses that facilitate the study of mental health and the components involved in it.

### 8.3. Limitations and Future Studies

The main limitations of this study are the following:

First of all, the lack of reliable data on some countries that have not been transparent in reporting information. This has made it difficult to obtain an accurate trend and the inclusion of studies along these lines. On the other hand, there is the lack of specific studies on certain vulnerable populations and the prevalence of mental pathologies.

This indicates that in future lines of research, it is advisable to analyze the data that have been published recently to better understand the determinants of the disease in these populations.

**Author Contributions:** Conceptualization: A.I.B.-V. and M.R.; methodology: A.I.B.-V. and M.R.; writing: A.I.B.-V., M.R., M.D.-G., S.U. and B.R.-E. All authors have read and agreed to the published version of the manuscript.

**Funding:** This research received no external funding.

**Institutional Review Board Statement:** Not applicable.

**Informed Consent Statement:** Not applicable.

**Data Availability Statement:** Not applicable.

**Conflicts of Interest:** The authors declare no conflict of interest.

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
