# Peer review of "Impact of COVID-19 Confinement on Mental Health in Youth and Vulnerable Populations: An Extensive Narrative Review"

_sustainability, doi:10.3390/su15043087_

Round 1
Reviewer 1 Report
The manuscript is well-written and interesting; however, I have included these gentle comments:
Gentle Comments and Typos
- Line 134: Please write the full form of "PTSD" “Post-traumatic stress disorder (PTSD)”.
- Line 184: ‘sleep patrons’ ‘sleep patterns’
Author Response
Dear Editor:
Please find enclosed the reviewed version of our manuscript to Sustainability entitled “Impact of COVID-19 Confinement on Diagnosis and Medication Use in Mood Disorders in Youth and Vulnerable Populations. An Extensive Narrative Review” (Sustainability-2145873). We would like to thank you for your considerations and the helpful comments of the reviewers. All the suggestions have been carefully attended. Please find below a point-by-point response to the reviewers.
We want to express our appreciation for taking the time and effort to provide such insightful guidance.
We hope that the manuscript in its present form meets all the requirements for publication in Sustainability.
Sincerely,
Manuel Reiriz/Ana I. Beltrán-Velasco
Nebrija University
C/ Santa Cruz de Marcenado, 27. Madrid. Spain.

Reviewer 2 Report
Dear Authors
This manuscript “Impact of COVID-19 Confinement on Diagnosis and Medication Use in Mood Disorders in Youth and Vulnerable Populations. An Extensive Narrative Review” was interesting, and authors should be applauded for doing a narrative review article.
This manuscript was well written, However, I have some suggestions that authors needs to include in the manuscript. Although narrative review are more descriptive which includes authors perspectives on a focused topic. Therefore, it is suggested that authors include two three sub-section in the conclusion section, and there are as follows:
8.1 Practical implications
8.2 Theoretical implications
8.3 Limitations and future studies
Author Response

(The authors gave the same response as above.)

Reviewer 3 Report
Many thanks for an interesting study, and the time and effort put into its preparation on such an important topic. I have the following comments:
In the title use colon (:) in place of full stop (.) after the word Populations.
Impact of COVID-19 Confinement on Diagnosis and Medication Use in Mood Disorders in Youth and Vulnerable Populations: An Extensive Narrative Review.
In abstract section, write few lines on the study outcomes based on the objectives of the study.
The title of the study reflects that ‘Diagnosis and Medication’ are two main variables in the review study. However, the objective written in the first line of the abstract has missed these two words. Objectives of the study should be aligned with the title of the study.
The same objective should also be written in the last para of the introduction section.
Major part of the paper is about the impact of covid-19 confinement on the metal health except point 3, i.e. Increased medication use. Therefore ‘Diagnosis and Medication’ can be removed from the title. (Even in conclusion part, nothing has been written on diagnosis and medication).
Heading 5 should be COVID-19 confinement and its impact on Vulnerable Population
Heading 5.2.3 should be ‘Other Mental Health Issues’
Be careful while writing the numerical. For example, in line 296, write 900000 in place of 900.000. In line 341, 58,8% and 82,3% should be written as 58.8% and 82.3%.
Bring uniformity while writing references. For example: In reference 2, you have mentioned et al. after 6 authors, whereas reference 6 is different.
Author Response

(The authors gave the same response as above.)

Reviewer 4 Report
This paper addresses an important topic, highlighting the impact of Covid-19 on people's mental health. I only have a few minor suggestions for the authors to consider and these are:
- It would be useful to consider amending the abstract to give the reader a better indication of the main arguments/findings being presented in the paper.
- It would be useful for the authors to define what vulnerable populations they will focus on or how they are defining vulnerability. For instance, people in prison are often considered a vulnerable population but are not discussed in this paper.
- There are some very long sentences in this paper that need to be shortened and referenced.
- On occasion, the phrasing is awkward and needs amending. If the authors first language is not English, it may be helpful to have someone whose first language is English to review the paper. I did not notice any typos or spelling errors but rather the occasional awkward expression.
- I think it would also strengthen the paper to acknowledge the limitations of a narrative review.
Author Response

(The authors gave the same response as above.)
